# Phytochemical and In Vitro Genotoxicity Studies of Standardized *Ficus deltoidea* var. *kunstleri* Aqueous Extract

**DOI:** 10.3390/plants10020343

**Published:** 2021-02-11

**Authors:** Hussin Muhammad, Maizatul Hasyima Omar, Elda Nurafnie Ibnu Rasid, Shazlan Noor Suhaimi, Farah Huda Mohkiar, Lau Mei Siu, Norizah Awang

**Affiliations:** Herbal Medicine Research Centre, Institute for Medical Research, National Institutes of Health, Ministry of Health, Malaysia, Level 5, Block C7, No. 1, Jalan Setia Murni U13/52, Seksyen U13, Setia Alam, Shah Alam 40170, Selangor, Malaysia; maizatul.hasyima@moh.gov.my (M.H.O.); elda@moh.gov.my (E.N.I.R.); shazlannoor@moh.gov.my (S.N.S.); farahhuda.m@moh.gov.my (F.H.M.); laums@moh.gov.my (L.M.S.); norizah@moh.gov.my (N.A.)

**Keywords:** *Ficus deltoidea*, genotoxicity, *Salmonella typhimurium*, alkaline comet assay, micronucleus

## Abstract

The present study was carried out to assess the genotoxicity potential of *Ficus deltoidea* var. *kunstleri* aqueous extract (FDAE) using standard in vitro assays. The DNA damage of V79B cells was measured using the alkaline comet assay treated at 0.1 mg/mL (IC10) and 0.3 mg/mL (IC25) of FDAE together with positive and negative controls. For in vitro micronucleus assay, the V79B cells were treated with FDAE at five different concentrations (5, 2.5, 1.25, 0.625, and 0.3125 mg/mL) with and without S9 mixture. The bacteria reverse mutation assay of FDAE was performed on *Salmonella typhimurium* strains TA98, 100, 1535, 1537, and *Escherichia coli* strain WP2*uvrA* using pre-incubation method in the presence or in the absence of an extrinsic metabolic system (S9 mixture). FDAE at 0.1 and 0.3 mg/mL significantly increased DNA damage in both comet tail and tail moment (*p* < 0.05). No significant changes were detected in the number of micronucleated cell when compared to control. Tested at the doses up to 5000 µg/plate, the FDAE did not increase the number of revertant colonies for all strains. In conclusion, further investigation needs to be conducted in animal model to confirm the non-genotoxicity activities of FDAE.

## 1. Introduction

Medicinal herbs have been utilized as a complementary and alternative medicine for many years in which it has gained widespread popularity throughout the world in treating diseases and improving healthy life in human-beings [1,2]. Herbs, herbal preparations, herbal materials, or finished herbal products are the herbal medicines that contain active ingredients [1]. Herbal products are generally not prepared according to the standard procedures of formulation as well as their evaluation of safety; hence, the herbal toxicity has become a major concern due to their wide use [3]. Furthermore, overutilization of natural materials without a proper toxicity data could potentially act as a carcinogen or mutagen [4]. Therefore, it is crucial to perform the safety assessment of herbal medicines particularly on their effects on genetic materials which may leads to cell mutation.

*Ficus deltoidea* (family: Moraceae) or locally known as “Mas Cotek” is one of the most common plants used in herbal based products development in Malaysia. However, it is native and widely distributed in other Southeast Asia countries including Thailand and Indonesia since the past few decades [5]. *F. deltoidea* is an epiphytic shrub which has been traditionally used for relieving headache, treating cold, rheumatism, and to strengthening the uterus after birth [6,7]. Most of the benefits are contributed by the flavonoids found in the plants. Scientifically, flavonoids have shown various pharmacological properties such as anti-carcinogenic, anti-mutagenic, anti-inflammatory, anti-viral, and anti-oxidant activities [8,9], though little is known about their safety and potentially detrimental toxic effects which might take place if taken in large doses for a short period or a long period of time [10]. Notwithstanding the widespread use of *F. deltoidea* plant in traditional medicine, limited data on genotoxicity effects particularly on variety *kunstleri* is available.

To evaluate the genotoxic potential of *F. deltoidea* aqueous extract (FDAE), in vitro testing through alkaline comet assay, micronucleus and *Salmonella typhimurium*/reverse mutation assay were performed in the present study.

## 2. Results

### 2.1. Phytochemical Analysis

The analysis of aqueous extract of *F. deltoidea* by reverse phase HPLC (RP-HPLC-ESI) revealed that flavonoids were the major class of phenolics in FDAE. Fifteen flavonoids were characterized. The obtained based peak chromatogram is illustrated in Figure 1. The phenolic compounds present in the FDAE are summarized in Table 1, with their formula, name, absorbance spectrum, calculated m/z, MS fragments, and migration.

### 2.2. MTS Cytotoxicity Assay

The potency of cell growth inhibition for FDAE on V79B cells was assayed. The cytotoxicity effect of the extract was measured after 24 h of exposure to a serial range of concentrations using MTS assay. The growth inhibitory effects were expressed as IC10 and IC25 (Figure 2). The toxic effect of FDAE exhibited in a dose-dependent manner and at a concentration of 5 mg/mL extract, the cell viability was 3.3 ± 4.1% relative to untreated controls. The growth inhibitory effect for IC10 and IC25 value were 0.1 and 0.3 mg/mL, respectively.

### 2.3. Alkaline Comet Assay

The DNA percentage in the comet tail and tail moment were used to measure the potential DNA damage caused by FDAE. The extract induced DNA damage by a significant increase in both % of DNA in tail and tail moment (*p* < 0.05) at concentrations of 0.1 mg/mL and 0.3 mg/mL, respectively, as shown in Figure 3.

### 2.4. In Vitro Micronucleus Assay

In the dose finding test, there were no precipitation of FDAE for all concentration observed in plates after treatment period. The FDAE was not cytotoxic at up to the highest concentration (5 mg/mL) in the presence and absence of S9 mixture, thus this dose was selected as the highest concentration for main experiment. Treatment with FDAE at 1.25, 2.5, and 5 mg/mL did not show a concentration-related increase in the percentage of micronucleated cells in both the absence and presence of metabolic activation compared to negative control. Figure 4 shows the percentage of micronucleated cells.

### 2.5. Salmonella typhimurium/Microsome Assay

A preliminary test was carried out in all tester strains and neither reductions of the number of the revertants or alterations of the auxotrophic bacterial grown were apparent up to the highest dose tested (5000 µg/plate). No dose-dependent increase in revertant colonies or bacterial toxicity were observed in the dose finding test at concentrations of FDAE up to 5000 μg/plate in the presence or absence of metabolic activation. The main experiment was performed in a similar manner at concentrations ranging from 313 to 5000 μg/plate. Data are summarized in Table 2 and Table 3. No bacterial toxicity was observed at any dose tested. The positive controls for each strain resulted in the expected increase in the number of revertant colonies.

## 3. Discussion

Despite the keen interest of traditional herbal medicine being used to treat various diseases, the toxicity and genotoxicity information of these herbal preparations are still lacking. Majority believes that products originated from natural sources are safe and they should devoid of toxicity. Therefore, an assessment of genotoxicity of leaves of *F. deltoidea* aqueous extract is required to ensure the safety of the *F. deltoidea*. In the current work, the mutagenic, and/or genotoxic of FDAE were studied with various assays.

Polyphenol-rich plants are often attributed with their potent antioxidant activities which are beneficial to human health. Flavonoids are phenolics found in the vegetables and plant-derived products such as flowers, fruits, roots bark, stems, grains, wine, and tea [11]. Knowledge on the flavonoids content of plant-based foods is paramount for better understanding in their role in plant physiology as well as human health. Identification of these compounds will determine their therapeutic value.

Our finding showed that flavonoids and phenolic compounds were identified in FDAE as reported in the previous study [12]. This study revealed FDAE consist of catechin and epicatechin, which are powerful antioxidant together with the presence of vitexin and apigenin, which have been demonstrated to have antioxidant with antigenotoxicity properties [13,14].

The presence of phenolic and flavonoid compounds has received a great attention on their free radical scavenging activities as a cellular defense mechanism to suppress the toxicity and genotoxicity of various mutagens in life organism. However, besides their beneficial effects, various studies have been reported about the cellular toxicity effects of flavonoids as a result of their pro-oxidant activities [15]. The cytotoxicity effects of FDAE were measured in V79B cell line by the MTS assay. Results of the present study clearly showed that the FDAE reduced the cell viability at up to the highest concentration in dose dependent manner. Similar cytotoxicity activity was observed in the methanolic extracts of *F. deltoidea* against human leukemic HL-60 cell lines as the cellular DNA fragmentation was revealed through microscopic evaluation on dead cells [16]. The presence of polyphenolic compounds in FDAE could influence cell function by modulating cell signaling, altering proliferation and inducing cytotoxicity in cells and requires further investigations.

The genotoxicity effect of FDAE was further evaluated at gene level based on DNA fragmentation from alkaline comet assay. This assay uses single-cell nucleus electrophoresis which can detect primary DNA damage expressed as single or double strand breaks [17]. It is based on the quantification of denatured DNA fragments that have migrated out from the cell nucleus during electrophoresis. The positive results obtained from this study indicated that FDAE could induce a direct genotoxic activity to produce DNA-lesions in V78B cells. The mechanisms by which FDAE may interact with DNA are difficult to predict. One possible cause contributing to the positive response observed in this assay could be the pro-oxidant activity of catechins found in the extract. It has been demonstrated that polyphenols, including catechins, can act as antioxidants and also as pro-oxidants under certain conditions [18], which induced oxidative damage in vitro studies [19,20,21]. Moreover, other findings showed that EGCG (epigallocatechin gallate), a predominant component of catechin preparations, induced chromosomal damage in WIL2-NS cells at 100 μmol/L [22]. However, whether polyphenols can act as pro-oxidants or antioxidants is likely dependent upon the relative redox-potentials of the in vitro culture conditions. A report by Ogura et al. (2008) [23] indicated that the pro-oxidant activity of catechins in in vitro culture was generated from the ample supply of dissolved oxygen. Other possible mechanism of action that cause DNA damage could be also due to the generation of reactive oxygen species [24] or the inhibition of scavenging enzymes against reactive oxygen species including catalase, superoxide dismutase and glutathione peroxidase [25] which require further investigations to confirm these activities.

In contrast, data from the *Salmonella typhimurium*/microsome assay provides no evidence of mutagenic potential as no increase of the number of revertant colonies when tested up to the highest dose (5000 µg/plate) over the negative control, either in the presence and the absence of extrinsic metabolic activation, S9. Standard mutagens (2-AA, NaN_3_, MMC, 2-NF and ICR-191), however, exhibited significant amplified number of revertant colonies in all strains. Similar finding was observed in TA98 and TA100 when tested with standardized methanolic extract of *F. deltoidea* [26]. Negative response was also observed in the study on green tea extract that rich with catechin compound at the highest concentration (5000 µg/plate). These data demonstrate no evidence of potential gene mutagenic effect under the conditions used in this test for FDAE.

To determine whether FDAE causes numerical and structural chromosome changes, an in vitro micronucleus test using V79B cells was conducted. In the present study, no significant increase in the number of micronuclei in any tested dose of FDAE compared with the negative control. These results are also consistent with the study findings of Ogura et al. (2008) [23] which assessed the potential genotoxic effect of a catechin rich tea preparation on ICR mice and Sprague Dawley rats using a bone marrow micronucleus assay. These results have shown that FDAE was not mutagenic in both the absence and presence of metabolic activation towards V79B cells.

## 4. Materials and Methods

### 4.1. Plant Materials and Extract Preparation

*F. deltoidea* dried leaves were purchased locally from Pahang, Malaysia. Leaves were dried in oven (Memmert, Germany) at 45 °C for 2 days and ground to a powdered form prior to extraction with water by boiling it for 1 h before assign through filter paper (Whatman, No.1). Filtrates collected were sprayed dried (Buchi, Switzerland) to form a powder that was further used in the experiments. The powdered aqueous extract of *F. deltoidea* (FDAE) were then analyzed by HPLC with absorbance, fluorescence and mass spectrometric detection to determine the major constituents. The HPLC system employed consisted of Surveyor gradient (Thermo Scientific, CA, USA), comprising of a pumping system, auto sampler, and degasser coupled with photodiode array absorbance (PDA) detector scanning from 200 to 700 nm controlled by Xcalibur software version 1.3. Separation was carried out using MAX-RP 4 µm, 250 mm × 4.6 mm C12 reverse phase column (Phenomenex, Torrance, CA, USA) maintained at 40 °C and eluted at a flow rate of 1.0 mL/min with 60 min gradient from 15 to 50% methanol in water containing 0.1% formic acid. After passing through the flow cell of the photodiode array and fluorescence detectors, the column eluate was split and 20% directed to an LCQ Duo mass spectrometer (Thermo-Finnigan) with an electrospray interface operating in full scan data dependent MS/MS mode from 150 to 1000 amu. (+)-Catechin and (−)-epicatechin were identified by fluorescence (FL) detector (Jasco FP-920) at the wavelengths (λ EX/ λ EM) 280/315 nm.

### 4.2. MTS Cytotoxicity Assay

#### 4.2.1. Cell Culture

V79B cell is a fibroblast-like morphology cell line originated from Chinese hamster lung. It was obtained from RIKEN Cell Bank, Japan (Resource No: RBRC-RCB2337). Cells were thawed and subcultured in Dulbecco’s Modified Eagle Medium (DMEM) supplemented with 10% Fetal Bovine Serum and 1% antibiotic/antimycotic solution and incubated in a humidified CO_2_ incubator at 37 ± 2 °C. Cells were washed with phosphate buffer solution and harvested by trypsinization using trypsin-EDTA 0.05% (*w/v*). The cells suspension was then centrifuged at 1000 rpm for 3 min. The pellet was re-suspended in the culture medium and the cells were counted using the Countess^TM^ Automated Cell Counter.

#### 4.2.2. Cytotoxicity Determination

Six treatment concentrations of FDAE (5, 2.5, 1.25, 0.625, 0.3125, and 0.15625 mg/mL) were prepared for the experiment. Culture medium from each well of 96-well plate was discarded and added with 200 µL extract at different concentrations and culture medium for negative control. Well without cells were used as blank. The plate was incubated for 24 ± 2 h at 37 ± 2 °C with 5 ± 1% CO_2_ in CO_2_ incubator. After 24 ± 2 h of incubation, 40 µL of MTS reagents was added into each well. The plate was then incubated for about 1 h at 37 ± 2 °C with 5 ± 1% CO_2_ in CO_2_ incubator. The plate was read at 492 nm using the ELISA Microplate reader. A concentration against percentage of cell viability graph was plotted. Two cytotoxic concentrations (IC10 and IC25) were selected for the Alkaline Comet assay.

### 4.3. Genotoxicity Assays

#### 4.3.1. Alkaline Comet Assay

All culture medium from each well of 96 well plates were discarded and washed with 2 mL of PBS. Two mL of FDAE (IC10 and IC25), negative control and positive control were pipetted into designated well of 6-well plate. The plate was incubated for 2 h at 37 ± 2 °C and 5 ± 1% CO_2_ in CO_2_ incubator. After 2 h of incubation, all culture media was discarded. Cells were washed, trypsinized with 1 mL of Trypsin-EDTA and 2 mL of culture medium to neutralize the reaction into each well. Cells were re-suspended and 1 mL transferred into three different microcentrifuge tubes and centrifuged at 2500 rpm at 4 °C for 5 min. The supernatant was discarded and 1 mL of cold PBS was added into each tube and centrifuged again at 2500 rpm at 4 °C for 5 min. The supernatant was discarded, then 80 µL of LMA (low melting agarose) was mixed with the pallet and re-suspended gently. The mixture was then pipetted onto the slide with 100 µL solidified NMA (normal melting agarose) and covered with cover slip. The agarose was allowed to solidify on ice for few minutes. The cover slip was removed and slides were immersed into lysis buffer solution containing 1% *v/v* Triton X-100 for 1 h at 2–8 °C. The slides were then transferred onto electrophoresis tank filled with buffer solution and left for 20 min to permit unwinding of DNA. After 20 min, the electrophoresis process was initiated (300 mA and 25 V) for another 20 min. The slides were transferred and washed with neutralization buffer solution every 5 min for 3 times. Then, 40 µL of ethidium bromide solution was pipette onto slides and covered with cover slips. The slides were kept overnight at 2–8 °C and observed under fluorescence microscope. Fifty single cells were analyzed for each slide using Tritek CometScore@ free software. Score for % DNA in tail and tail moment was collected and transferred into Alkaline Comet Assay Excel Template.

#### 4.3.2. In Vitro Micronucleus Assay

The assay was performed according to OECD 487 in vitro Mammalian Cell Micronucleus [27] with minor modifications. Two mL of V79B cells suspension (5 × 10^4^ cells/mL) were dispensed into designated well of 6-well plate. Cells were incubated for 24 ± 2 h at 37 ± 2 °C and 5% CO_2_ in incubator. After 24 h, the subconfluency and the morphology of the cells were verified. V79B cells were washed with 1 mL of PBS. Then, 5 mL of culture media containing either 1.25, 2.5, and 5 mg/mL FDAE, and negative control or positive control (colchicine-without S9 activation and cyclophosphamide monohydrate-with S9 activation, respectively) were added into each well. The plates were incubated at 37 ± 2 °C and 5% CO_2_ for 3 h. After 3 h of treatment period, treatment culture media was discarded and cells were washed twice using PBS. Then, 5 mL of normal culture media was added into each well and plate was incubated at 37 ± 2 °C and 5 ± 1% CO_2_ for 21 h. After incubation, the culture media was replaced with 2 mL of PBS-EDTA. The cells were re-suspended to get single cells. The cells suspension was then centrifuged at 1500 rpm. The cells pellet were re-suspended with 2 mL of KCl hypotonic solution and the cell again were centrifuged at 1500 rpm for 5 min. The cells pellet was fixed using Carnoy’s fixative solution. 10 µL cells were drop and mounted on pre-warmed frosted slides. The cells were stained using 10 µL of 20 µg/mL acridine orange and observed under fluorescence microscope at 20–40× magnification. The images of 2000 cells were captured and scored.

#### 4.3.3. Statistical Analysis

SPSS version 16.0 was used for the statistical analysis. Data was analysed using one-way ANOVA and presented in mean and standard error of the mean. Statistical significance was considered with *p* value < 0.05.

### 4.4. Salmonella typhimurium/Microsome Assay

#### 4.4.1. Bacterial Strains and Metabolic Activation System (S9 Mixture)

*Salmonella typhimurium* strains and *Escherichia coli* strain were purchased from the Molecular Toxicology Incorporated (Moltox), United States. The bacterial strains were stored as frozen stock cultures at −80 °C. The cultures were preserved with Dimethyl sulfoxide (DMSO) in 1 mL of broth culture. For metabolic activation system, a cofactor-supplemented post-mitochondrial fraction (S9) was used. The S9 was prepared from the livers of male Sprague Dawley rats induced with phenobarbital and 5,6-benzoflavone.

#### 4.4.2. Positive Control Mutagen

2-aminoanthracene (2-AA), Mitomycin C (MMC), ICR-191, Benzo(a)pyrene (BP), Sodium azide (SA) and 2-nitrofluorene (2-NF) were from Sigma-Aldrich, St Louis Missouri United States.

#### 4.4.3. Mutagenicity Assay

This study was conducted in accordance with Method 471, Bacterial Reverse Mutation Test (Adopted 21 July 1997), OECD Guideline for testing of chemicals. The *Salmonella typhimurium*/microsome assay was performed by the standard pre-incubation method with and without addition of an extrinsic metabolic activation system (S9 mixture). The test strains (TA98, TA 100, TA1535, TA1537, and WP2*uvrA*) were cultured in Nutrient Broth no. 2 and incubated at 37 °C for 6–9 h. Basically, 100 μL of an overnight grown culture (containing approximately 1–2 × 10^9^ bacterial cells per mL) was added into culture tubes which contained 100 μL of FDAE (5000, 2500, 1250, 625, 313, and 0 μg) or standard mutagens (positive control) or sterile distilled water (negative control), and 500 μL of phosphate buffer (without S9) or 500 μL of S9 mixture. The mixture was incubated for 20 min at 35 ± 2 °C with shaking speed at 100 rpm in the shaker water bath. Two mL of overlay agar was then added into each culture tubes followed by vigorous mixing and poured on the surface of a minimal agar. The plate was incubated for 48 h at 37 ± 2 °C. All plates were then checked for the presence of the background lawn and compared to the negative control group plates. Numbers of revertant bacterial colonies were counted and compared with those in negative and positive control plates. Every experiment was carried out in triplicate.

#### 4.4.4. Statistical Analysis

SPSS version 16.0 was used for the statistical analysis. Data was analysed using one-way ANOVA and presented in mean and standard deviation. Statistical significance was considered with *p* value < 0.05.

## 5. Conclusions

Herbal derived medicinal products have become popular to treat several diseases despite the scarce information on the toxicity of certain compounds which may detrimental to human health. In this work, we showed that FDAE was not mutagenic in both in vitro micronucleus and bacterial reverse mutation assays. We suggested that the anti-mutagenic activity could be attributed to the presence of flavonoids with good antioxidant and antigenotoxicity in FDEA. In contrast, DNA damaged was observed when tested using alkaline comet assay towards V79B cells under the in vitro condition. The cellular activities of FDAE may be modulated by a variety of interactions between mutagenic and protective compounds, which may result to a different response pattern. Therefore, further studies using animal model (in vivo) should be conducted to better characterize the mechanism of action and to confirm the genotoxicity effects of FDAE under physiological conditions.

## Figures and Tables

**Figure 1 plants-10-00343-f001:**
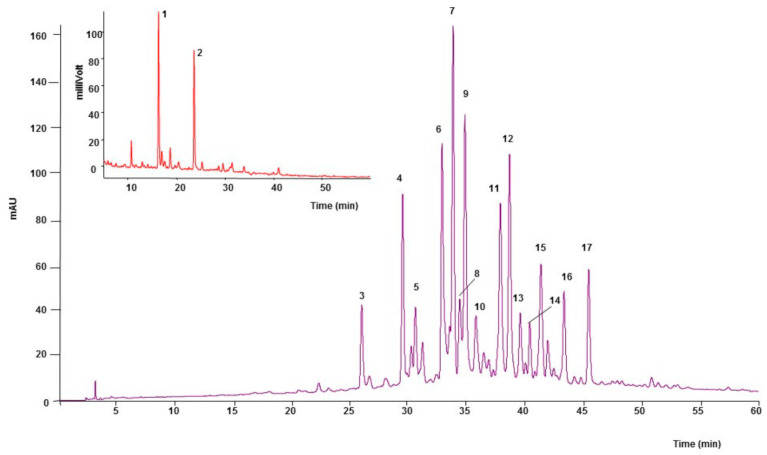
HPLC-ESI/MS profile of *F. deltoidea* aqueous extract (FDAE), analyzed by gradient phase at 365 nm (purple) and florescence detector (red).

**Figure 2 plants-10-00343-f002:**
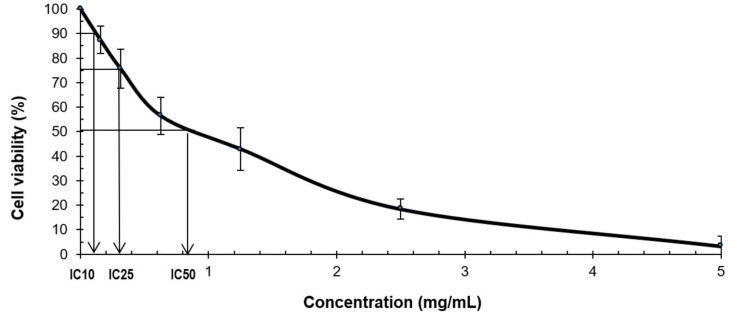
Cytotoxic effect of FDAE was demonstrated following 24 ± 2 h treatment on V79B cells. Data are expressed as percentage of viability compared to negative control (without any treatment). IC10: concentration of Test Item that produce 10% inhibition of cell viability. IC25: concentration of Test Item that produce 25% inhibition of cell viability. IC50: concentration of Test Item that produce 50% inhibition of cell viability.

**Figure 3 plants-10-00343-f003:**
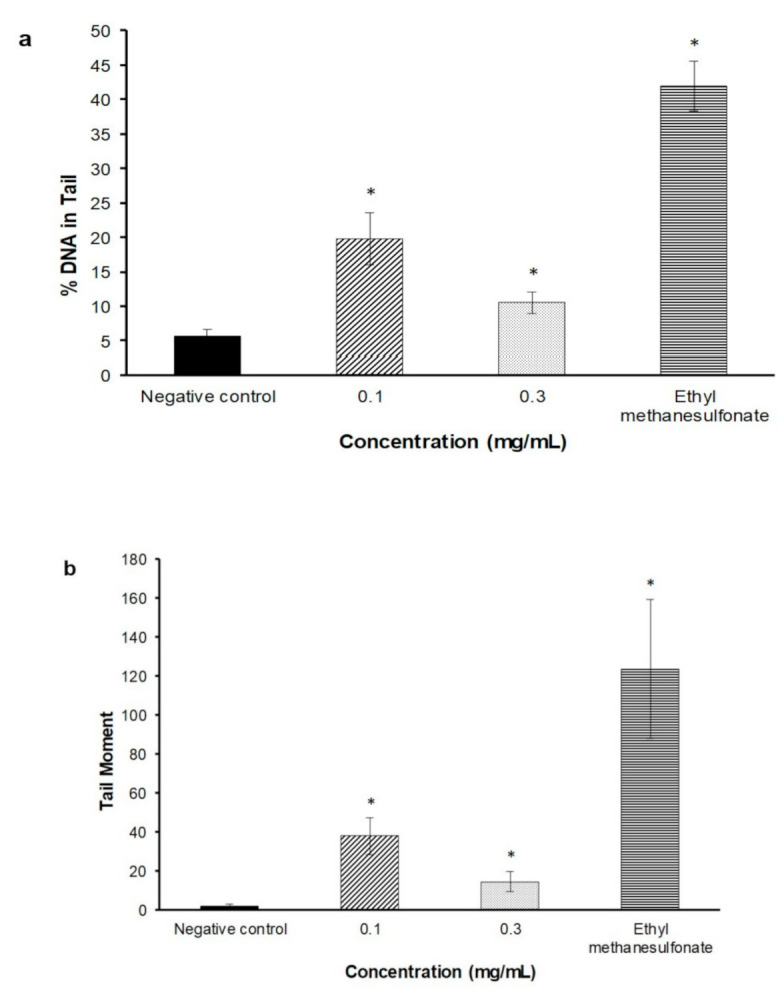
The DNA damage induction in V79B cells measured by alkaline comet assay after cells were treated either with FDAE or positive control (Ethyl methanesulfonate) for 2 h. DNA damage is expressed as (**a**): % DNA in tail and (**b**): tail moment. Data was expressed as mean ± SEM. * is where *p* value is < 0.05 as compared to negative control.

**Figure 4 plants-10-00343-f004:**
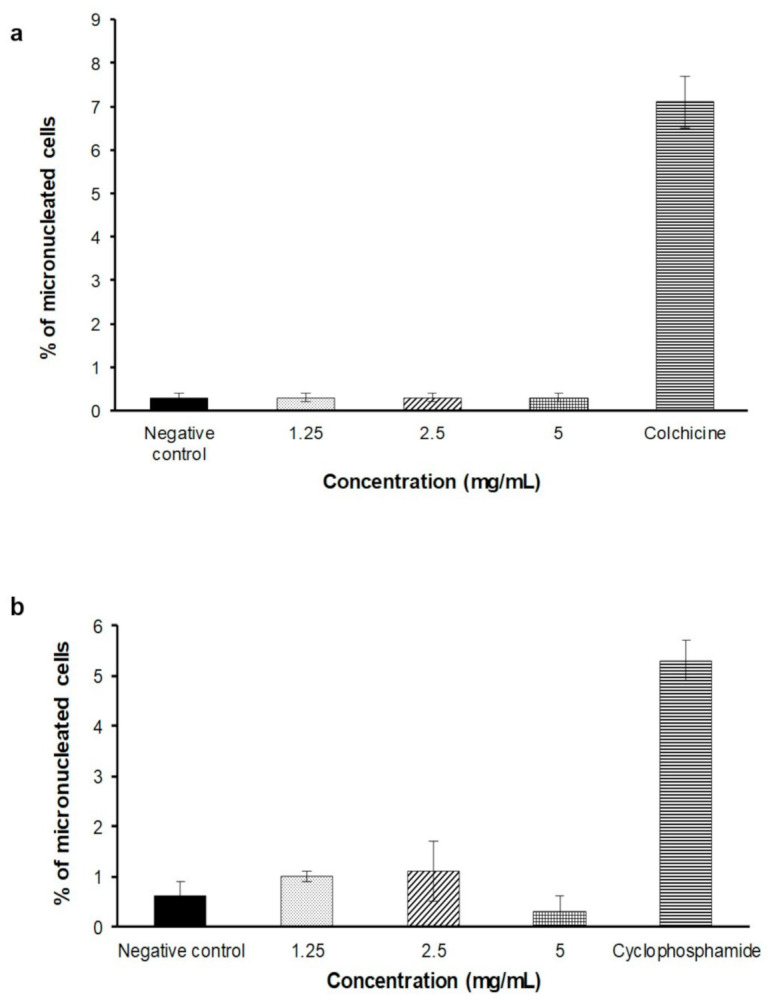
Percentage of micronucleated cells induced by different concentrations of FDAE or positive controls treated on V79B cells for 3 h with (**a**) and without (**b**) the presence of S9 metabolic activation. Data was expressed as mean ± SEM.

**Table 1 plants-10-00343-t001:** Tentative identification of phenolic compounds in *F. deltoidea* extract determined by HPLC-ESI.

Peak	t_R_	λ_max_	Compound	[M-H]^−^ (m/z)	MS2 Fragments Ions (m/z)
1	16.2	280	(+)-Catechin	289	245, 179
2	23.5	280	(−)-Epicatechin	289	245, 205, 179
3	25.9	350	Luteolin-6-8-*C*-diglucoside (Lucenin-2)	609	489, 519, 399
4	29.4	340	Apigenin-6, 8-*C*-diglucoside (Vicenin-2)	593	473, 503, 353
5	30.5	345	Unidentified	517	385, 205, 222
6	32.8	335	Apigenin-6-*C*-ara-8-*C*-glucoside (Isoschaftoside)	563	473, 443, 503
7	33.7	335	Apigenin-6-*C*-glu-8-*C*-arabinoside (Schaftoside)	563	473, 503, 443
8	34.3	345	Luteolin 6-*C*-glucoside (Isoorientin)	447	327, 357, 369
9	34.8	325	Unidentified	565	444, 474, 443
10	35.7	335	Luteolin-8-*C*-diglucoside (Orientin)	447	327, 429, 357
11	37.8	335	Apigenin -8-*C*-glucoside (Vitexin)	431	311, 341, 283
12	38.6	335	Apigenin-6-*C*-pent-8-*C*-glucoside	563	443, 473, 353
13	39.5	335	Apigenin-6*-C-*pentosyl*-*8*-C-*pentoside	533	443, 473, 515
14	40.3	335	Apigenin-6*-C-*pentosyl*-*8*-C-*pentoside	533	443, 473, 515
15	41.3	345	Apigenin 6-*C*-glucoside (Isovitexin)	431	311, 341, 413
16	43.3	335	Unidentified	535	443, 474, 516
17	45.4	335	Chrysin-6-8-*C*-diglucoside	577	457, 487, 353

**Table 2 plants-10-00343-t002:** Mutagenicity testing of FDAE in the *Salmonella typhimurium*/microsome assay without S9 metabolic activation on TA100, TA1535, TA98, TA1537, and WP2*uvrA* tester strains.

Dose (μg/plate)	Number of Revertants (Mean ± SD)				
	Base-Pair Substitution Type			Frameshift Type	
	TA100	TA1535	WP2*uvrA*	TA98	TA1537
0	136 ± 10	13 ± 2	57 ± 5	21 ± 4	8 ± 2
313	145 ± 12	14 ± 1	53 ± 3	21 ± 1	9 ± 2
625	135 ± 5	10 ± 2	58 ± 4	20 ± 3	9 ± 2
1250	130 ± 8	11 ± 3	48 ± 4	19 ± 1	11 ± 2
2500	134 ± 11	10 ± 1	55 ± 2	23 ± 4	10 ± 2
5000	166 ± 7	12 ± 1	53 ± 3	25 ± 2	11 ± 3
Positive control	627 ± 6	558 ± 6	180 ± 7	539 ± 4	608 ± 4

Values are mean ± SD of 3 plates. Dose 0-negative control: 100 µL pure water; Positive control: for TA100 and TA1535, NaN_3_ (0.5 µg/plate); WP2*uvrA*, MMC (1 µg/plate); TA98, 2-NF (1 µg/plate) and TA1537, ICR-191 (1 µg/plate).

**Table 3 plants-10-00343-t003:** Mutagenicity testing of FDAE in the *Salmonella typhimurium*/microsome assay with S9 metabolic activation on TA100, TA1535, TA98, TA1537 and WP2*uvrA* tester strains.

Dose (μg/plate)	Number of Revertants (Mean ± SD)				
	Base-Pair Substitution Type			Frameshift Type	
	TA100	TA1535	WP2*uvrA*	TA98	TA1537
0	162 ± 6	17 ± 3	60 ± 4	24 ± 4	10 ± 1
313	151 ± 10	15 ± 2	63 ± 4	30 ± 4	11 ± 2
625	145 ± 12	14 ± 2	63 ± 2	28 ± 2	12 ± 2
1250	158 ± 5	15 ± 2	53 ± 4	32 ± 3	10 ± 2
2500	142 ± 3	14 ± 1	57 ± 5	28 ± 2	12 ± 2
5000	149 ± 6	14 ± 3	62 ± 8	29 ± 2	11 ± 1
Positive control	519 ± 4	165 ± 4	192 ± 4	232 ± 6	183 ± 3

Values are mean ± SD of 3 plates. Dose 0-negative control: 100 µL pure water; Positive control: 2AA for all tester strains, TA100 (1 µg/plate); TA1535 (2 µg/plate); WP2*uvrA* (10 µg/plate); TA98 (0.5 µg/plate) and TA1537 (2 µg/plate).

## Data Availability

Not applicable.

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
