# Peer review of "Phytochemical and In Vitro Genotoxicity Studies of Standardized *Ficus deltoidea* var. *kunstleri* Aqueous Extract"

_plants, 2021, doi:10.3390/plants10020343_

Round 1

Reviewer 1 Report

Genotoxicity assay of chemicals/biochemicals is a very important issue, which directly related to the understanding of inherent toxic effect on human beings. This topic has attracted great attentions in the field of biomedicine and healthcare, molecular toxicology, as well as sustainable social development. In this study, the authors evaluated the genotoxicity potential of Ficus deltoidea var kunstleri aqueous extract (FDAE) using standard in vitro assays. By using the alkaline comet assay, the DNA damage of V79B cells (treated at 0.1 mg/mL and 0.3 mg/mL of FDAE) was measured together with positive and negative controls. The V79B cells were also treated with FDAE at 5 different concentrations (5, 2.5, 1.25, 0.625 and 0.3125 mg/mL) with and without S9 mixture in vitro micronucleus assay. The bacteria reverse mutation assay of FDAE was performed on S. typhimurium strains TA98, 100, 1535, 1537 and E. coli strain WP2uvrA using pre-incubation method in the presence or in the absence of an extrinsic metabolic system, increased DNA damage in both comet tail and tail. This work provided some useful information for understanding the genotoxicity of mixture of natural product extracts, there are several issues need to be addressed before consideration:

  1. The MTS assay of FDAE should be performed with a positive control such as flavones or other phenolic compounds in V79B cells, which would be better for the comparison of apparent cytotoxicity.
  2. The authors performed MTS assay of FDAE, how about the cell membrane disruption properties of FDAE? Sometimes the cytotoxicity was induced by ROS-mediated membrane lipid peroxidation, the reviewer suggests that Lactate Dehydrogenase (LDH) activity assay of FDAE need to be performed.
  3. The electrophoresis profiles of comet assay of the DNA damage under various concentrations of FDAE need to be photographed and added into the manuscript.
  4. Why use the DNA alkylation agent ethyl methanesulfonate as the positive control? From the FDAE in Table 1, it seems that all the compounds are phenolic derivatives and no alkylation agents involved, the reviewer suggested that the positive control need to be changed to a phenolic compound.
  5. The mechanism of FDAE-induced DNA damage is not clear, it seems originated from ROS-induced DNA damage. The reviewer suggested that, after the addition of various amount of FDAE, the intracellular ROS concentration/activity need to be measured by DCFH-DA (2’-7’dichlorofluorescin diacetate) ROS Assay kit in the

Author Response

1. The MTS assay of FDAE should be performed with a positive control such as flavones or other phenolic compounds in V79B cells, which would be better for the comparison of apparent cytotoxicity.

The cytotoxicity assay was conducted as pre-experiment to aid in dose selection for genotoxicity study that to be used in the alkaline comet assay. Only negative control was used as a comparison to the FDAE and no positive control. However, to determine the genotoxicity activity of FDAE, positive control, ethyl methanesulfonate (EMS) was used in the study as it is mutagenic to plants and animals and recommended by the guidelines. The suggestion of using the flavones or other phenolic compounds especially with known genotoxicity effects as positive control will be considered in the future study.

2. The authors performed MTS assay of FDAE, how about the cell membrane disruption properties of FDAE? Sometimes the cytotoxicity was induced by ROS-mediated membrane lipid peroxidation, the reviewer suggests that Lactate Dehydrogenase (LDH) activity assay of FDAE need to be performed.

Our main objective performing the cytotoxicity was to determine cell inhibition of FDAE on the selected cell, V79B. In this case, we determine CC10 and CC25 based on their cell viability determined by their mitochondrial activity that were measured to select concentrations to proceed with the alkaline comet assay. In determination of probable mechanism of cell cytotoxicity related to ROS-mediated membrane lipid proxidation, the suggestion to include the Lactate Dehydrogenase (LDH) activity will be considered for future study.

3. The electrophoresis profiles of comet assay of the DNA damage under various concentrations of FDAE need to be photographed and added into the manuscript.

Thank you for your suggestion. We will consider adding the photographs to the manuscript. 

4. Why use the DNA alkylation agent ethyl methanesulfonate as the positive control? From the FDAE in Table 1, it seems that all the compounds are phenolic derivatives and no alkylation agents involved, the reviewer suggested that the positive control need to be changed to a phenolic compound.

Ethyl methanesulfonate is a well-established genotoxic agent that has been used extensively as model compound in experimental work to establish the responsiveness of the test system like V79B. Ethyl methanesulfonate induces DNA damage by a direct mechanism, acting as a monofunctional ethylating agent. The suggestion of using the flavones or other phenolic compounds especially with known genotoxicity effects as positive control will be considered in the future study.

5. The mechanism of FDAE-induced DNA damage is not clear, it seems originated from ROS-induced DNA damage. The reviewer suggested that, after the addition of various amount of FDAE, the intracellular ROS concentration/activity need to be measured by DCFH-DA (2’-7’dichlorofluorescin diacetate) ROS Assay kit in the

Agreed with the reviewer. The genotoxic effects of FDAE could be due to direct or indirect effects which may activate the cellular DNA-damage response. A suggestion to perform the ROS assay will consider to be performed in future experiments.

Reviewer 2 Report

This is a very interesting study about the genotoxicity potential of Ficus deltoidea aqueous extract using standard in vitro assays, because, despite the prevalent use of this plant as a food and medicine, the toxicity of F. deltoidea has not been fully explored.

The manuscript is well written and it highlights important scientific point. The abstract presents very clear the objectives of the study. The results can be useful for research in the field. However, there are some minor observations:

Line 220. Explain the reason why you used these concentrations and not others.

Line 288. Please specify briefly (in parentheses) what are the minor modifications about you are talking.

At Material and method section the details about the statistical method used is missing!

In my opinion, in the future studies should be also determined the content of heavy metals in the leaves of F. deltoidea.

Author Response

Line 220. Explain the reason why you used these concentrations and not others.

In the previous study conducted by Mat Akhir et al., (2011) on the cytotoxicity effects of Ficus deltoidea aqueous extract tested on Human Ovarian Carcinoma cell line has shown that the dose that cause 50% inhibitory of cell viability (IC50) was 224.39 µg/ml. Therefore, in our present study, the dose range finding was performed with the objective to find the IC50 when the V79B cell treated with FDAE. The highest dose, 5mg/ml was used to observe the cytotoxicity activity of FDAE and from there, concentration selected for the alkaline comet assay should cover a range from that 50% producing cytotoxicity to little or no cytotoxicity. These concentrations range cover approximately half log interval between each concentration.

Reference : Mat Akhir, N., Chua, L., Abdul Majid, F., & Sarmidi, M. (2011). Cytotoxicity of Aqueous and Ethanolic Extracts of Ficus deltoidea on Human Ovarian Carcinoma Cell Line. Journal of Advances in Medicine and Medical Research1(4), 397-409. https://doi.org/10.9734/BJMMR/2011/507

Line 288. Please specify briefly (in parentheses) what are the minor modifications about you are talking.

In the OECD487 guidelines has mentioned about the requirements of temperature and CO2 level required for plates and cell culture incubation shall be at 37oC and 5%, respectively. However, in our study, the temperature and CO2 level tolerance for incubator were set as 37 ±2oC and 5±1% as all equipment runs with tolerance.

At Material and method section the details about the statistical method used is missing!

Thank you for your comment. We will add the statistical method used in this study. 

In my opinion, in the future studies should be also determined the content of heavy metals in the leaves of F. deltoidea.

Thank you for your suggestion. We will take into consideration of the suggestion for our future study.

Reviewer 3 Report

Authors present an analysis on the genotoxic effects of an aqueous extract of Ficus deltoidea var. kunstleri.

Although the theme is interesting, some methodological and argumentative criticalities must be corrected by the authors.

1) The authors state that Ficus deltoidea kunstler does not exhibit genotoxic effects, but only report results on the aqueous extract and not on an at least hydroalcoholic extract that may contain components with genotoxic activity. This aspect should be discussed by the authors.

2) Figure 2 is unclear. In my opinion, it is not clear what the cytotoxic effect of the extract is compared to the control. I suggest using clearer graphics.

3) It is unclear why the authors use 0.1 and 0.3 mg / mL for their genotoxicity tests. Authors should explain better.

4) The authors should better explain why they use V79B cells

5) The most critical observation concerns the absence of a statistical analysis. On the basis of which statistical approach were the significances identified? Authors need to justify the identified significances.

Author Response

1) The authors state that Ficus deltoidea kunstler does not exhibit genotoxic effects, but only report results on the aqueous extract and not on an at least hydroalcoholic extract that may contain components with genotoxic activity. This aspect should be discussed by the authors.

We will add “ Results from the AMES test was in agreement with the study of Farsi et al., (2013) where the standardized methanolic extract of F.deltoidea leaves extract did not induce gene mutation in both the with or without metabolic activation S9”. Apart from that, we study the aqueous extract as in this part of research, we only focus on herbal consumption of FDAE as how they are prepared traditionally or consumed.

2) Figure 2 is unclear. In my opinion, it is not clear what the cytotoxic effect of the extract is compared to the control. I suggest using clearer graphics.

Graph 2 demonstrated the cell growth inhibition of V79B to determine CC10 and CC25 as reference to proceed with the comet assay. We will adjust accordingly and present the graph to illustrate the inhibition of FDAE on V79B. 

3) It is unclear why the authors use 0.1 and 0.3 mg / mL for their genotoxicity tests. Authors should explain better.

The 0.1 and 0.3mg/ml concentrations used in the study was based on the results of cytotoxicity study of FDAE. It has been recommended to use the range of concentrations that produce the cytotoxicity and to little or no cytotoxicity. From the cytotoxicity study it was found that the concentration that cause 10 % and 25% inhibition of cell viability were 0.1 and 0.3mg/ml respectively. Therefore, these concentrations were used in the genotoxicity test as it was expected to see the effects of these concentrations on the DNA damage which was conducted using the alkaline comet assay

4) The authors should better explain why they use V79B cells

Chinese hamster lung fibroblast (V79) has been widely used in studies of DNA damage and DNA repair. In fact, this cell has been recommended by the OECD guidelines for the testing of chemicals as it is used for mutagenesis and mammalian cell genetics. The morphology of V79B differs from that of V79 cells as V79B cells are more efficient in stable transfections. Several mutants with increased sensitivity to different genotoxic agents have been isolated from V79B.

5) The most critical observation concerns the absence of a statistical analysis. On the basis of which statistical approach were the significances identified? Authors need to justify the identified significances.

We will add the statistical approach and analysis method in the manuscript.

Round 2

Reviewer 1 Report

The reviewer suggests that the revised manuscript could be considered for acceptance at the current statement.

Author Response

The reviewer suggests that the revised manuscript could be considered for acceptance at the current statement.

Thank you for your comment. 

Reviewer 3 Report

The authors have finalized their paper following some indications suggested by the Reviewers.   I am partially satisfied with the corrections and additions made by the authors. Data on cytotoxicity are not yet clear.   In my opinion, the authors should add some references such as:   

1) Misbah, H., Aziz, A.A. & Aminudin, N. Antidiabetic and antioxidant properties of Ficus deltoidea fruit extracts and fractions. BMC Complement Altern Med 13, 118 (2013); 
2) Mastinu A, et al. Gamma-oryzanol Prevents LPS-induced Brain Inflammation and Cognitive Impairment in Adult Mice. Nutrients. 2019;
3) Hamidun Bunawan, Noriha Mat Amin, Siti Noraini Bunawan, Syarul Nataqain Baharum, Normah Mohd Noor, "Ficus deltoidea Jack: A Review on Its Phytochemical and Pharmacological Importance", Evidence-Based Complementary and Alternative Medicine, vol. 2014, Article ID 902734, 8 pages, 2014. https://doi.org/10.1155/2014/902734    

Author Response

The authors have finalized their paper following some indications suggested by the Reviewers. I am partially satisfied with the corrections and additions made by the authors. Data on cytotoxicity are not yet clear. In my opinion, the authors should add some references such as: 

The cytotoxicity study was conducted to establish IC10 and IC25. Therefore, the graph presented as growth inhibitory effect with increasing concentration of FDAE. From this graph, we obtain the inhibition concentration.

With regards to the reference, the comments for this are as follows:-

  1. Misbah, H., Aziz, A.A. & Aminudin, N. Antidiabetic and antioxidant properties of Ficus deltoidea fruit extracts and fractions. BMC Complement Altern Med 13, 118 (2013); 

We have added to the manuscript in the discussion section, reference No. 12

  1. Mastinu A, et al. Gamma-oryzanol Prevents LPS-induced Brain Inflammation and Cognitive Impairment in Adult Mice. Nutrients. 2019;

We do apologize as we can't add the reference for now as we think it is not applicable to the current study.

  1. Hamidun Bunawan, Noriha Mat Amin, Siti Noraini Bunawan, Syarul Nataqain Baharum, Normah Mohd Noor, "Ficus deltoidea Jack: A Review on Its Phytochemical and Pharmacological Importance", Evidence-Based Complementary and Alternative Medicine, vol. 2014, Article ID 902734, 8 pages, 2014. https://doi.org/10.1155/2014/902734    

We have already cited this paper which is reference no.7.